# The Role of Sensory Innervation in Homeostatic and Injury-Induced Corneal Epithelial Renewal

**DOI:** 10.3390/ijms241612615

**Published:** 2023-08-09

**Authors:** Konstantin Feinberg, Kiana Tajdaran, Kaveh Mirmoeini, Simeon C. Daeschler, Mario A. Henriquez, Katelyn E. Stevens, Chilando M. Mulenga, Arif Hussain, Pedram Hamrah, Asim Ali, Tessa Gordon, Gregory H. Borschel

**Affiliations:** 1Department of Surgery, Indiana University School of Medicine, Indianapolis, IN 46202, USA; 2Program in Neurosciences and Mental Health, Hospital for Sick Children, Toronto, ON M5G 0A4, Canada; 3Department of Hand, Plastic and Reconstructive Surgery, Burn Center, BG Trauma Hospital, Department of Plastic and Hand Surgery, University of Heidelberg, 67071 Ludwigshafen, Germany; 4Cornea Service, New England Eye Center, Tufts Medical Center, Department of Ophthalmology, Tufts University School of Medicine, Boston, MA 02111, USA; 5Center for Translational Ocular Immunology, Department of Ophthalmology, Tufts Medical Center, Tufts University School of Medicine, Boston, MA 02111, USA; 6Department of Ophthalmology and Vision Sciences, Hospital for Sick Children, Toronto, ON M5G 1X8, Canada; 7Department of Ophthalmology and Vision Sciences, University of Toronto, Toronto, ON M5T 3A9, Canada; 8Department of Ophthalmology, Indiana University School of Medicine, Indianapolis, IN 46202, USA

**Keywords:** limbus, limbal niche, limbal stem cells, Schwann cells, cornea, corneal innervation, corneal denervation, neurotrophic keratopathy, corneal neurotization, FK-506, tacrolimus, nerve growth factor, cenegermin

## Abstract

The cornea is the window through which we see the world. Corneal clarity is required for vision, and blindness occurs when the cornea becomes opaque. The cornea is covered by unique transparent epithelial cells that serve as an outermost cellular barrier bordering between the cornea and the external environment. Corneal sensory nerves protect the cornea from injury by triggering tearing and blink reflexes, and are also thought to regulate corneal epithelial renewal via unknown mechanism(s). When protective corneal sensory innervation is absent due to infection, trauma, intracranial tumors, surgery, or congenital causes, permanent blindness results from repetitive epithelial microtraumas and failure to heal. The condition is termed neurotrophic keratopathy (NK), with an incidence of 5:10,000 people worldwide. In this report, we review the currently available therapeutic solutions for NK and discuss the progress in our understanding of how the sensory nerves induce corneal epithelial renewal.

## 1. Introduction

Corneal clarity is required for vision, and blindness occurs when it becomes opaque. Similar to the epidermis of the skin, the superficial corneal epithelium is continually sloughed off and replaced as it shields the eye from external insults [1]. Corneal epithelial renewal [2,3] depends on the activity of limbal stem cells (LSCs) [4,5,6,7] that are located in the basal epithelium of the limbus, surrounding the cornea [6,8]. The limbus, which contains the Palisades of Vogt, forms the transition zone between the cornea and conjunctiva [1,9]. In the limbal niche, the LSCs interact with mesenchymal stromal cells (MSCs; are also addressed as limbal niche cells) and T cells that play a critical role in the maintenance and activity of the LSCs [10,11,12]. Normally, the LSCs reside in a growth-arrested or slow-cycling state [13,14,15,16], exhibit morphological characteristics of stem cells, and express genes associated with asymmetric cell division [17,18,19,20,21]. During homeostatic epithelial renewal or after injury, the LSC progeny differentiate to transient amplifying cells (TACs) that migrate radially to the central cornea where they differentiate further into epithelial cells, thereby replenishing the corneal epithelium [11,12,22,23,24,25,26,27,28,29,30,31,32] (Figure 1).

The cornea is innervated by sensory nerves that protect it from injury by activating tearing and blink reflexes [33]. The axons in the basal epithelial layer of the limbus run adjacent to the LSCs, and their free nerve endings contact epithelial cells [34]. There is a growing volume of evidence suggesting a critical role of corneal sensory innervation in regulating its epithelial renewal via stimulating the activity of LSCs [35,36,37,38,39,40,41,42,43,44,45,46,47,48,49,50,51]. When protective sensory innervation is lost after infection, trauma, intracranial tumors or ocular surgery, or when innervation fails to develop in congenital cases, permanent blindness results from repetitive microtraumas that induce epithelial breakdown, ulceration, scarring, and opacification of the cornea [36,38,52,53] (Figure 2). This condition, known as neurotrophic keratopathy (NK), affects nearly 5/10,000 people [54,55]. Patients with NK develop persistent breakdown of the epithelium [36,38,52,53] and poor healing [53,56,57,58,59] that, in turn, inevitably cause scarring and opacification of the cornea [54,60,61,62,63]. The field’s prevailing conceptual model is that corneal epithelial wound healing and maintenance depend on sensory axons providing direct trophic stimulation of LSCs [35,36,37,38,39,40,41,42,43,44,45,46,47,48,49,50,51]. Two conceptually different therapeutic approaches for NK have evolved: (i) a novel surgical procedure termed “minimally invasive corneal neurotization” (MICN), designed to restore or provide corneal sensory innervation, and (ii) development of a topical treatment in attempt to maintain or recover a physiological trophic milieu that is lost following pathological corneal denervation [64,65,66]. Despite progress in treatments for NK, nearly 30–40% of NK patients fail to respond to either of the available treatments. Therefore, NK remains one of the most difficult ophthalmic conditions to treat, and continues to represent a leading cause of corneal blindness worldwide [54].

As with all peripheral nerve fibers, the corneal axons interact with Schwann cells (SCs) [67,68]. In addition to their role supporting axonal maintenance and conductivity, SCs in the nerve terminals also play a key role in tissue regeneration. After skin wounding or digit tip amputation, mature SCs dedifferentiate into SC precursors that produce cytokines and growth factors which mediate recovery [69,70]. There is an abundance of myelinating and non-myelinating SCs in the corneal limbus that are closely associated with LSCs [67,68,71]. Recently, Mirmoeini et al. [71] suggested that SCs serve as the key cellular mediator of corneal innervation-dependent epithelial renewal, acting via trophic regulation of LSCs.

Here, we summarize (i) the progress in our understanding of the mechanism of corneal sensory innervation-dependent epithelial renewal; (ii) the role of SCs in regulation of the process, particularly; (iii) known and potentially new intercellular interactions involved in the regulation of LSCs, and of the limbal niche activity during homeostatic and injury-induced corneal epithelial renewal; (iv) the role and possible mechanisms of NGF activity in this process; and (v) available and potential new pharmacological and surgical therapeutic approaches to treat NK.

## 2. Corneal Sensory Innervation

The sensory innervation of the cornea protects against environmental microtraumas by triggering tearing and blink reflexes [33]. Given the crucial role of vision in survival and the necessity of corneal transparency for vision, the cornea is the most highly sensory nerve-innervated region of the body’s surface [72]. Trigeminal nerve fibers enter the cornea via the suprachoroid space and extend in a radial pattern parallel to the corneal surface. Stemming from their annular plexus at the periphery, these nerves terminate as free nerve endings in the limbal region, forming the limbal plexus [34]. The axons in the basal epithelial layer of the limbus run adjacent to the LSCs, with their free nerve endings making contact with epithelial cells [34] (Figure 1). Nerve fibers from the deeper stroma proceed towards the epithelium to form the subepithelial nerve plexus, and then ascend to Bowman’s layer and the basement membrane, dividing into thinner fibers that form the sub-basal nerve plexus. Finally, fibers from the latter plexus penetrate the epithelial layer (Figure 1) forming whorl-like or vortex patterns on the corneal epithelium’s surface [34,73] (Figure 3).

Most epithelial nerve fibers are polymodal non-myelinated C fibers that convey sharp pain in response to microtraumas and chemical stimulation. The remaining nerve endings are myelinated Aδ fibers that are responsive to cold stimuli [34,72]. These nerve fibers retain their myelination at the limbus and the peripheral part of the stroma, but gradually lose the myelin sheath as they approach the center of the cornea [34].

Besides protecting the cornea from injury, sensory nerve fibers with their accompanying cells likely play a crucial role in LSC proliferation and survival [35,36,37,38,59,64,74], which suggests that corneal nerves are critical regulators of epithelial renewal. Corneal nerves-secreted signaling molecules, including neurotrophins and neurotransmitters, as well as non-neuronal trophic factors, are likely regulators of LSC activity. This would explain why in the condition of NK, a pathological deficit in corneal sensory innervation, interferes with the epithelial regenerative capacity and its progressive degeneration [36,37,75,76].

## 3. Schwann Cells and Innervation-Dependent Corneal Epithelial Renewal

Multiple studies addressed the importance of the limbal niche in regulating LSC maintenance and activity. MSCs and T cells are the key cellular components of the niche, regulating self-renewal, proliferation, and dedifferentiation of LSCs [10,11,12]. Following limbal injury, the niche is also capable of inducing the dedifferentiation of terminally committed epithelial cells to replenish the pool of LSCs [23]. Paradoxically, despite the well-known critical role of corneal innervation, no attempts had been made until recently to find the linking factor(s) that connect between the corneal nerves and the activity of the limbal niche.

One previously prevalent school of thought argued that corneal epithelial stem cells differentiate vertically to replenish the upper layers of the epithelial cells [77]. More recent reports, using single-cell RNA sequencing (scRNA-seq) and lineage tracing approaches in mice, unequivocally showed that LSCs are restricted to the limbal niche, which is also the mostly richly innervated part of the cornea [11,12,22,23,24] (Figure 1). These studies revealed a centripetal migration of differentiated LCS progeny during corneal epithelial renewal. This newer conceptual model suggests a different mechanism for the regulation of LSCs, which are distant from the wounded area, unlike in other tissues, such as skin, in which the progenitor cells differentiate vertically [69,70].

In the limbus, axons of the cornea-innervating nerves are associated with myelinating and non-myelinating SCs in a 1:1 relation, and are situated in close vicinity to LSCs (Figure 1) [71]. In the peripheral nerve system, in addition to the well-studied role of SCs in axonal trophic support and electrophysiological activity, SCs in nerve terminals play a key role in regulating tissue regeneration and homeostatic stem cells activity [69,70,78]. In mammalian wounded skin, mature SCs dedifferentiate into SC precursors that produce cytokines and growth factors, which mediate wound closure [70]. Similarly to skin wounds, during regeneration of a digit tip, SC precursors at the wound bed secrete PDGF-AA and oncostatin M (OSM) to induce proliferation of mesenchymal cells [69]. While the neurotrophic and neurophysiological role of SCs in the cornea has been addressed [67,68], their potential role in regulating the function of the limbal niche cells during epithelial renewal had not been investigated until recently. Unlike in skin and digit tips, where nerve axons-dissociated Sox2-positive SC precursors occupy the wound bed [69,70], in the cornea, axons-associated Sox10-positive mature SCs together with LSCs are localized in the limbus, distant from the central corneal epithelium [71]. This observation would support the hypothesis of a conceptually different mechanism of SC-mediated regulation of tissue renewal activity that is more similar to that found in the bone marrow, in which mature non-myelinating SCs at nerve terminals secrete TGF-b to induce hibernation of hematopoietic stem cells [78]. In this regard, Mirmoeini et al. demonstrated that experimental corneal denervation in rats, by stereotactic electrocautery of the ophthalmomaxillary branch of the trigeminal nerve [71,74], caused a rapid and complete loss of SCs. Additionally, induced local genetic ablation of SCs in mice corneas completely prevented recovery of a wounded epithelium [71]. Like clinically anesthetic corneas, SC-ablated corneas developed the NK phenotype, characterized by corneal opacification and ulceration [71]. Notably, unlike in corneal denervation, ablation of SCs did not affect the blink reflex or sensory innervation, which remained normal throughout the course of the experiment [71]. These findings refuted the notion that corneal nerve terminals govern corneal epithelial renewal; rather, corneal nerve-associated SCs play a much greater role. Considering the trophic inter-dependence between SCs and neurons [79,80,81], in Mirmoeini et al.’s suggested model, corneal axons function as SC “carriers” [71].

The comprehensive analysis performed by Mirmoeini et al. provides important insight into the regulatory complexity of innervation-dependent corneal epithelial renewal. The breakthrough findings provide the missing linking factor by suggesting SCs as a new cellular component of the limbal niche that mediates corneal innervation-dependent epithelial renewal [71].

## 4. Trophic Regulation of Sensory Innervation-Mediated Corneal Epithelial Renewal

Deficiencies in the signaling molecules, including neurotrophins and neurotransmitters, is a possible, if not a likely contributor to the demise of the epithelium after corneal denervation, because the deficiency results in a loss of regenerative capacity and progressive epithelial degeneration [36,37,75,76]. Concentrations of acetylcholine and the neuropeptide substance P in the corneal epithelium and tears have been shown to decline after corneal denervation [36,82,83]. In several in vitro experiments, pro-proliferative and cell migration-promoting effects of neuropeptides or neurotrophins, such as substance P in combination with IGF-1, have been observed in corneal epithelial cells and limbal stem cells [84,85,86]. Among non-neuronal origin trophic factors found in the corneal epithelium, ciliary neurotrophic factor (CNTF), transforming growth factor-β (TGF-β), platelet-derived growth factors (PDGFs), and NGF have been implicated in the regulation of corneal epithelial renewal [87,88,89,90,91,92]. Among these trophic factors, NGF is the only agent that has demonstrated a positive effect on corneal healing and maintenance in clinical studies.

New insight to the mechanistic understanding of the regulation of LSCs during corneal epithelial renewal arrived from a recently reported comparative scRNA-seq analysis of dissociated corneal limbi that were harvested from healthy, de-epithelialized healing and denervated corneas [71]. A complex regulatory trophic communication between limbal cell populations was proposed on the basis of the altered gene expression in the pathological conditions [71]. This model of paracrine interactions suggests that NGF acts synergistically with other trophic factors, including CNTF, PDGF-α, and TGF-β, locally expressed by SCs and MSCs in the limbus, to regulate the activity of LSCs. Moreover, the observed upregulation of *NGF* expression, particularly by SCs following de-epithelialization [71] suggests direct involvement of NGF signaling in the epithelial healing process. Whilst further research is required to validate the gene expression-based prediction, the progress in our understanding of the trophic interactions during innervation-dependent corneal epithelial renewal that is mediated by SCs should lead to development of a pharmacological treatment for NK. This treatment should compensate for the changes in the LSCs’ trophic milieu following deficits in the innervation of the cornea.

## 5. Currently Available Pharmacological Approaches to Treat NK

The current nonsurgical (i.e., conservative) treatments for NK include autologous serum topical applications [93], topical nerve growth factor (see below), as well as lubricating agents, such as artificial tears. Less expensive growth factors, such as topical insulin (1 U/mL applied three times per day) have also been used to accelerate healing of epithelial defects in patients with NK [94,95]. Others have used scleral lenses to slow the progression of NK. Surgical options have historically relied on tarsorrhaphy (suturing the eyelids together) to reduce the area of corneal surface exposed, and hence reducing evaporative losses and protecting the ocular surface. These methods help reduce the incidence of corneal ulceration in NK; however, many patients will eventually sustain a corneal ulcer despite these treatments. In most instances in which epithelial defects are intentionally created (e.g., photorefractive keratectomy), corneal epithelial wound healing progresses to full healing of the defect. However, when innervation is impaired for any reason (iatrogenic injury, diabetes, or other causes), then wound healing may be delayed, leading to chronic ulceration. To accelerate wound healing, clinicians often prescribe topical antibiotics, such as levofloxacin, in combination with topical corticosteroids [96]. Incomplete reinnervation can lead to a decline in corneal sensation, which, while fortunately rare, can be a cause of morbidity in some patients undergoing cornea procedures.

### Nerve Growth Factor (NGF)

Thus far, recombinant human (rhNGF; Cenegermin, Dompé, Milan, Italy) remains the only clinically approved topical treatment for NK [66,97]. However, despite its low toxicity and evidence for NGF’s stimulation of LSC activity [98] and corneal epithelial healing in human NK cases [50,99,100], NGF treatment of NK presents remaining concerns. Firstly, the treatment requires extremely high doses, prolonged treatment duration, and dosing every two hours. Secondly, NGF treatment fails in over 30% of patients, especially in those with more severe NK [66,97]. These observations suggest that NGF alone may not be sufficient to fully compensate for a lack of innervation in NK patients. Ironically, despite being clinically approved, neither the cellular source nor the mechanism of rhNGF’s activity in the cornea are known. Defining the mechanism could improve the treatment’s efficiency and reduce its exceptionally high cost: an 8-week therapy cycle comes with a median out-of-pocket patient cost of USD 5791, while the total gross drug cost for 2410 Medicare beneficiaries in 2019–2020 was USD 287 million [101] (an average cost of USD 119,000 per patient).

In preclinical in vivo studies, the topical application of NGF significantly accelerated the healing of epithelial defects in physiologically innervated corneas, and conversely, topical application of NGF antibodies reduced the rate of healing [49,102]. Placebo-controlled clinical studies that evaluated the therapeutic effect of topically applied NGF in patients with the corneal lesions of NK found significantly faster epithelial regeneration compared to the placebo group. There was a concomitant significant increase in corneal sensitivity, which suggested that the cornea had become reinnervated and that the reinnervation may be induced by the applied NGF [48,103].

NGF significantly increases the expression, axonal transport, and secretion of certain neuropeptides such as substance P in sensory neurons, suggesting a possible pathophysiological connection between the observed effects of the trophic factor and neurotransmitters [104]. The detection of NGF and its corresponding high-affinity receptor tropomyosin kinase A (TrkA) in corneal epithelial cells and LSCs in humans and murine species suggests partial autocrine or paracrine effects of NGF in the epithelium and LSCs [48,49]. In fact, NGF induced formation of LSC colonies in vitro [98]. NGF is well-documented as a potent neurotrophin that induces growth and regeneration of sensory neurons [105,106,107,108,109,110]. The axons of sensory neurons extend along an NGF gradient towards high NGF concentrations [111,112], which could be particularly relevant in the central corneal epithelium, where Schwann cells and the endoneurium, important guiding structures for sprouting axons, are absent [113,114]. Instead, corneal epithelial cells may function as surrogate Schwann cells to provide the epithelial neurites with required trophic support [115].

The expression of TrkA by both corneal sensory neurons and non-neuronal cells, and the local expression of other trophic factors rather than NGF suggested to be involved in corneal epithelial renewal, raise two possible explanations for the observed inconsistencies in the efficacy of NGF treatment in NK. Firstly, as NGF is the major effector of survival and growth of sensory and sympathetic neurons via activation of the TrkA receptor [116,117,118], treatment with NGF could facilitate corneal innervation, thereby indirectly improving corneal epithelial renewal. In this case, the treatment’s efficacy would depend on the pre-treatment level of innervation and/or the axonal growth potential. The latter could be limited in some congenital cases, in which impaired developmental sensory innervation is the original cause of the disease’s development [119]. Secondly, additional trophic factors that act synergistically with NGF may be required for activation of the LSCs to mediate innervation-dependent corneal epithelial renewal.

With reference to the pre-treatment level of corneal innervation, confocal microscopic observations in cenegermin-treated patients with NK demonstrated significant improvements in corneal innervation density [120]. On the other hand, there is limited, if any, effect of the treatment in patients with a completely anesthetic cornea [66]. Although no clear-cut correlation has been observed between the extent of corneal innervation and NGF treatment efficacy, these observations support the hypothesis that the main effect of NGF treatment is the induced growth of the corneal sensory axons that thereby improve corneal clarity in patients with NK. However, NGF signaling may also induce corneal epithelial renewal directly. Upregulation of *NGF* expression, particularly by SCs, following de-epithelialization [71], suggests direct involvement of NGF signaling, acting together with other trophic factors to regulate the epithelial renewal.

## 6. Strategies to Promote Corneal Reinnervation

### 6.1. Potential Treatment with Tacrolimus

Tacrolimus, a clinically approved immunosuppressant commonly used for various ophthalmic conditions [121], including allergic keratoconjunctivitis [122] and corneal transplantation [123], has been shown to promote axon regrowth in vitro [124,125,126,127] and axonal regeneration following nerve injury in vivo [128,129,130,131,132]. The direct neurotrophic effect of tacrolimus is mediated by the chaperone-like FK506 binding protein (FKBP52) [125,127,133,134], which forms heterocomplexes with the 90 kDa heat-shock protein (Hsp90) and its co-chaperone p23 [134] in the neuronal nucleus. Injured neurons redistribute this complex to the growth cones of regenerating neurites upon cellular contact with tacrolimus, promoting their accelerated regeneration [134]. FKBP52 also mediates neuronal growth cone guidance in response to attractive and repulsive chemotactic signals [135]. The systemic delivery of tacrolimus accelerates axonal regeneration in vivo by 12% to 16% [136,137]. Sustained local delivery of low-dose tacrolimus directly at the repair site of peripheral nerves increases the number of regenerating nerve fibers and accelerates their rate of regeneration [138,139,140]. Daeschler et al. adapted this approach, namely the sustained local delivery of the drug from a biodegradable drug delivery system, to promote reinnervation of denervated corneas in rats [65]. The density of regenerated nerve fibers in the central cornea was found to be significantly higher four weeks post-denervation, as compared to non-treated control rats [65].

Given the well-documented clinical safety record of tacrolimus in ophthalmic applications [141,142,143] as well as in other indications [144,145,146,147,148], this therapeutic approach could be seamlessly implemented in clinical trials. Furthermore, although a specialized drug delivery system designed for the sustained topical delivery of tacrolimus may not be necessary for preliminary clinical investigations, tacrolimus eye drops may be considered for self-administration during the day, albeit with a mean surface residence time of over 1.5 h [149]. It remains to be seen whether this frequency of dosing is adequate for maintaining therapeutic levels of tacrolimus in the cornea, and future investigations will shed light on this topic. An alternative strategy could involve implementing this approach in patients undergoing corneal neurotization, which would confer the added scientific advantages of a precisely standardized procedure with a known number of transferred donor nerve fibers, established via intraoperative nerve biopsies. Since this procedure entails a defined distance for the nerve regeneration and the entry points for the nerve fibers into the cornea, this may allow nerve fiber tracking using in vivo confocal microscopy. Hence, in patients afflicted with corneal denervation and NK, sustained and topical delivery of tacrolimus to the ocular surface holds promise as a viable and readily implementable therapeutic approach for promoting corneal nerve fiber regeneration.

### 6.2. Corneal Neurotization

Clinically, corneal “neurotization” surgery improves corneal wound healing and corneal innervation in patients with anesthetic corneas [64]. Neurotization entails harvest and placement of healthy functional nerves into a previously denervated tissue to regain motor or sensory function. This approach was first reported in the peer-reviewed literature by Terzis et al., who used direct neurotization from the contralateral supratrochlear (STN) and supraorbital nerves (SON) [150]. These nerves were transferred directly into the denervated cornea’s limbal area, which provided corneal sensation and enabled corneal healing in patients with anesthetic corneas. Later modifications of this procedure, which utilized nerve grafts to reduce invasiveness and allow for more distant sensory nerve sources, were similarly successful and made the procedure feasible for bilaterally affected patients and congenital patients with more widespread facial denervation [151] (Figure 4). Additionally, vision was improved in these patients due to healing of corneal opacifications [64,152]. In a recent meta-analysis, nerve-grafted patients experienced slightly greater sensory outcomes, which may be due to the increased axon density within grafts compared to distal ends of STN or SON [153]. However, the results are similar between direct and indirect corneal neurotization, and therefore the approach chosen varies by individual patient characteristics and surgeon preference. In vivo confocal microscopy has demonstrated improvements in corneal epithelium morphology and nerve ingrowth as early as three months post-operatively [154,155]. Representing a major paradigm shift, corneal neurotization enables patients to undergo corneal transplantation to further improve vision, as the transplants would previously re-opacify in patients with NK in the absence of innervation [54].

Corneal neurotization is an effective method for restoring corneal sensation for most patients suffering from corneal nerve fiber loss and NK [64,150]. However, this invasive procedure is limited by its availability, and requires a highly specialized surgical team. Although not always completely effective in treating NK, the use of this surgical treatment supports the tantalizing hypothesis that corneal epithelial wound healing and maintenance depend on certain cells within nerves, or products of those cells, interacting with LSCs to regulate these epithelial renewal-dependent processes.

## 7. Summary and Discussion

Corneal clarity, which is critical for vision, is achieved by protective and trophic activity of the corneal sensory innervation. The sensory nerves guarantee corneal clarity by (i) triggering blink and tearing reflexes, thereby preventing corneal wounding, and (ii) are thought to trophically stimulate LSCs activity, thereby inducing homeostatic and injury-induced epithelial renewal. In the pathological condition of NK due to impaired corneal sensory innervation, a repetitively wounded cornea fails to heal, leading to its opacification and, eventually, inevitable blindness. There are three clinically adapted and developing therapeutic approaches: (1) topical administration of recombinant trophic factors or factors-containing fluids that are aimed to re-establish the physiological trophic milieu, compromised in the innervation deficit conditions, thereby inducing healing of the epithelial lesion; and/or (2) topical administration of neurotrophins or chemical neurotrophic compounds to support corneal re-innervation; or (3) performing of corneal neurotization—surgically re-establishing corneal sensory innervation by introducing donor nerves into a denervated cornea. Among the available clinically approved topical agents, NGF is addressed as the mostly promising and potent. However, it is not active in 30% of NK cases, and the treatment suffers from substantial limitations due to the frequent and high dosage required, and its extremally high cost [50,99,100]. Corneal neurotization, although successful in general, fails to induce recovery of normal sensation in conditions of limited neuronal regenerative capacity, which often leads to recurrent ulceration [152]. In this review, we discussed the role of sensory innervation in corneal epithelial renewal, the progress in our current understanding of the process, and how these findings could potentially be implemented in clinics. In particular, the discovery of SCs as a key component of the process and, presumably as a trophic regulator of LSC activity [71]; moreover, its biological significance has a high therapeutic potential in the treatment of NK. Once identified, the corneal SC-derived recombinant topical trophic factors could either be administrated alone to support the activity of LSCs, or in combinations with NGF that would facilitate corneal re-innervation. Moreover, SCs themselves could be considered as a therapeutic target. Exogenous topical Neuregulin 1 Type I [79,80,156], by preventing denervation-caused SC withdrawal [71], could both induce corneal re-innervation and stimulate LSCs. The discovery of locally-released tacrolimus as a potent neurotrophin also has a high therapeutic potential to induce or accelerate corneal re-innervation, particularly in clinical cases when a spontaneous re-innervation capacity is preserved. Finally, the presented above potential therapeutic approaches could be applied alone, or could supplement surgical approaches such as corneal transplantation and/or neurotization, in dependence of the severity and/or original reason for the corneal sensory innervation deficit.

## 8. Methods of Literature Review

This manuscript consolidates the group’s clinical and research experience and findings, as well as reported findings of other groups studying fundamental regulatory aspects of corneal epithelial renewal and the clinical aspects of NK etiology and treatment. The literature summarized in this manuscript represents peer reviewed scientific and clinical reports that used the key words listed on the title page.

## Figures and Tables

**Figure 1 ijms-24-12615-f001:**
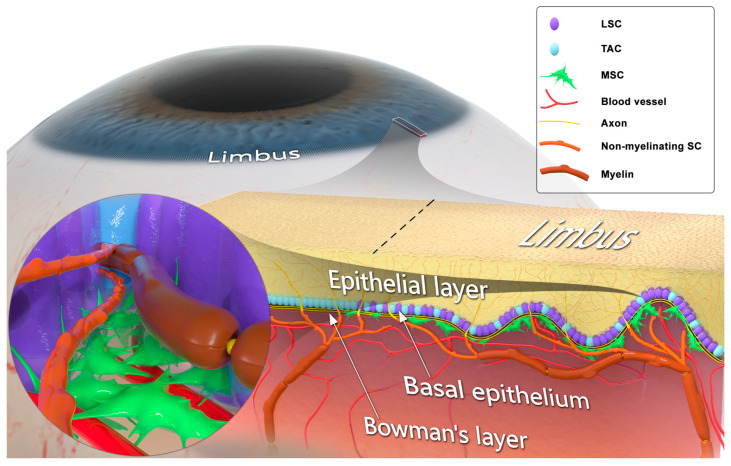
Structural and cellular composition of the limbal niche. The corneoscleral limbus contains limbal epithelial crypts, forming the limbal niche (or limbus). Limbal stem cells (LSCs) differentiate into transient amplifying cells (TACs) and, together with TACs, form the basal epithelial cellular layer. LSCs lie in close contact with mesenchymal stromal cells (MSCs), myelinating Schwann cells (SCs), ensheathing Aδ fibers, and non-myelinating SCs associated with C fibers. LSC, MSC, and SC populations engage in juxtaparacrine molecular crosstalk. The limbal MSCs are attached to the basement membrane, interact with blood vessels, and their projections pass through the basement membrane in contact with LSCs. SCs associated with unmyelinated sensory axons penetrate the basement membrane and extend neurites, which terminate in the epithelial surface.

**Figure 2 ijms-24-12615-f002:**
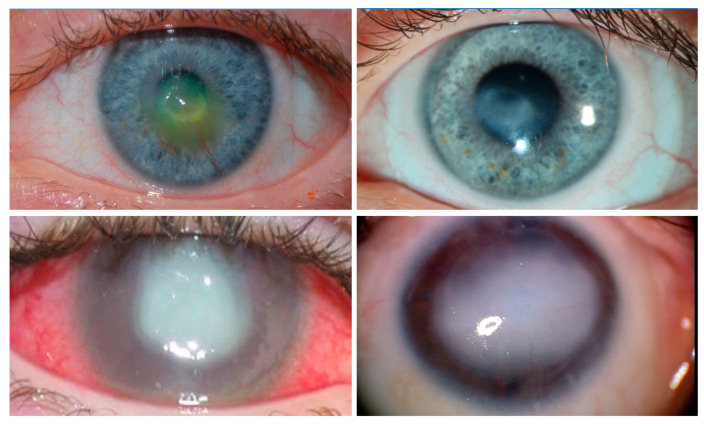
Clinical images of neurotrophic keratopathy (NK). Clinical appearance of anesthetic corneas, showing central corneal opacity with overlying epitheliopathy. Conjunctival injection, corneal neovascularization, and corneal stromal scar with overlying epithelial defect are demonstrated. (**Top left**): A cornea with a shallow corneal ulcer. The clinician has applied fluorescein, which indicates the location of the de-epithelialized corneal stroma. (**Top right**): Same patient as (**Top left**). The ulcerated area has progressed onwards to scarring as the epithelial defect heals. (**Bottom left**): A patient with severe NK leading to corneal perforation and infection. (**Bottom right**): A patient with recurrent ulcerations leading to total opacification of the cornea. All four of these patients were treated with corneal neurotization.

**Figure 3 ijms-24-12615-f003:**
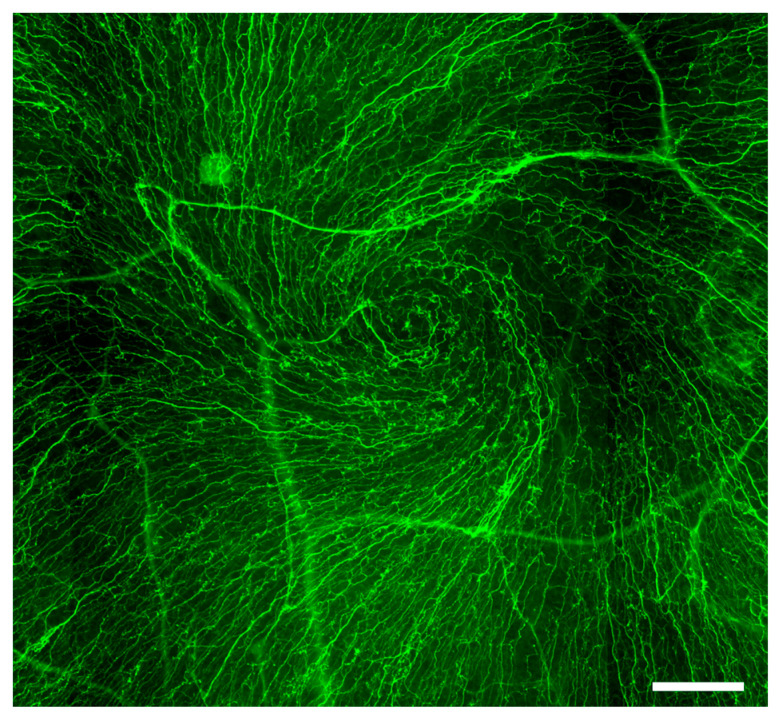
Whorl-like or vortex pattern of the corneal epithelial nerves. Immunofluorescent image of βIII-tubulin-stained whole-mount rat cornea demonstrates neurites of the corneal sensory axons forming a whorl-like or vortex pattern on the corneal epithelium’s surface. Scale bar: 100 μm.

**Figure 4 ijms-24-12615-f004:**
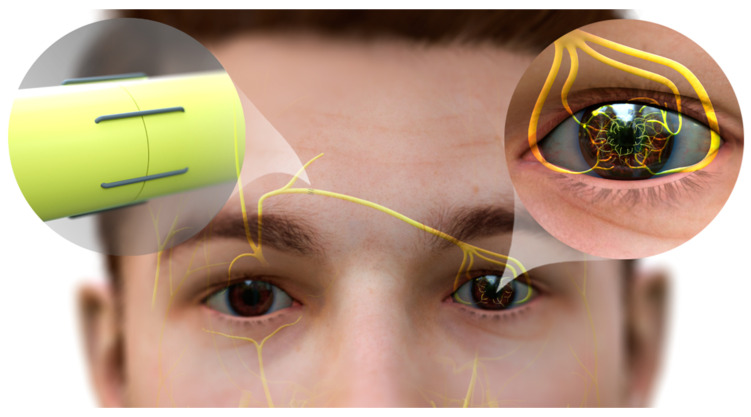
Neurotization of human cornea. The end of the sural nerve graft is coapted to the divided donor supraorbital nerve, permitting fibers to regenerate into the nerve graft. The nerve graft is passed subcutaneously from this incision into the contralateral upper lid counterincision. The graft is then passed into a subconjunctival plane and separated into its component fascicles, which are then tunneled and secured individually around the limbus. Axons of the donor nerve emerge from the graft, and grow into the corneal stroma and into the overlying epithelium. While most corneal neurotization patients gain sensation post-neurotization, many do not achieve the highest possible level of sensation as measured by standard Cochet–Bonnet esthesiometry (CBA, a variable length nylon monofilament). Those patients that achieved less than 50 mm of CBA (of maximum 60 mm) were not as protected against recurrent corneal ulceration post-operatively as those who achieved 60 mm on CBA [152]. This observation suggests that corneal neurotization, while revolutionary, may not be sufficient to provide full corneal sensation in many cases, leaving room for assistance from topical therapeutics.

## Data Availability

Data sharing not applicable.

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
