# Peer review of "The Role of Sensory Innervation in Homeostatic and Injury-Induced Corneal Epithelial Renewal"

_ijms, 2023, doi:10.3390/ijms241612615_

Round 1

Reviewer 1 Report

The scope of the IJMS is "providing an advanced forum for biochemistry, molecular and cell biology, molecular biophysics, molecular medicine, and all aspects of molecular research in chemistry". 

The MS is focused on the function of sensory innervation in the pathology and physiology of the cornea.

The MS should emphasize the under-mechanism of sensory innervation on the cornea, such as:

1. The Nerve-cornea junction,

2. the neurotrophic factors on the cornea,

3. the involved signaling pathway,

...

I suggest transferring to another Journal.

Author Response

We thank the reviewer for the valuable comments. Our goal in this manuscript is to review the progress in our understanding of (i) the role and the mechanism of sensory innervation in corneal epithelial renewal, with emphasis on the trophic rather than electrophysiological activity of the nerves (ii) the pathological condition of neurotrophic keratopathy resulted from deficit in the corneal innervation, and (iii) the existing and potential therapeutic approaches to treat neurotrophic keratopathy. The manuscript briefly summarizes structural aspect of the corneal sensory innervation, the list of previously known and newly discovered neurotrophins and non-neuronal trophic factors, potentially involved in the regulation of the corneal epithelial renewal, and the discussion potential intercellular interactions involved, with an emphasis on the new discovery of Schwann cells as a key cellular regulator of the process. The revised version contains critical structural changes that provide the information delivery in more deductive and logical way. A chapter with summary and discussion of the key points of the manuscript was added.

Reviewer 2 Report

In the current manuscript entitled “The Role of Sensory Innervation in Homeostatic and Injury-Induced Corneal Epithelial Renewal”, Konstantin Feinberg and collaborators have reviewed the currently available therapeutic solutions for neurotrophic keratopathy and discussed the progress by which the trigeminal sensory nerves induce corneal epithelial renewal.

The article is well written and summarizes the currently available and possible future therapeutic approaches to treat NK, the role of Schwann cells in regulation of corneal epithelial renewal, the possible mechanisms of NGF activity in regulation of LSCs and the new intercellular interactions that are potentially involved in the regulation of LSCs and of the limbal niche activity.

Author Response

We thank the reviewer for the valuable comments. Although the reviewer has expressed no major concerns, in the revised version we introduced critical structural changes that provide the information delivery in more deductive and logical way. A chapter with summary and discussion of the key points of the manuscript has been added.

Reviewer 3 Report

  1. Throughout the text, use the justify option of text alignment (align the text evenly between the margins).
  2. Line 4 – two commas after „Simeon C. Daeschler 2,3“.
  3. Line 23 – In the abstract section, try to rearrange the text so as not to repeat the word “cornea” so often.
  4. Line 52 – Figure 1. Text is broken by the figure itself.
  5. Authors are advised to have a gradation of the topics described - eg. to start the review from conservative and continue with invasive treatment options to achieve a more fluent reading.
  6. I suggest adding a short Method of literature review section, describing the process of data collection, databases and keywords used.
  7. I recommend adding a small description of present conservative (autologous serum, lubrication agents, etc.) and surgical (tarso-conjunctival flap) supportive treatment options for NK.
  8. Line 96 – additionally to already described options for epithelial healing as nicergoline, I suggest highlighting that other less expensive drugs have proven to be effective, namely topical insulin.
  9. Add discussion and conclusion to the text - the way it currently ends is very abrupt. You should add these sections to assist the reader in making conclusions from this review.

Minor editing required. 

Author Response

Response

We thank the reviewer for the valuable comments. Below are per point responses:

  1. The text was aligned accordingly
  2. The second comma was removed
  3. The use of the word “cornea” was reasonably minimized
  4. The figure was re-positioned

5. The text was re-arranged in more deductive and logical way

  1. The suggested section were added
  2. The suggested paragraph was added
  3. The suggested paragraph was added
  4. A summarizing chapter was added

Reviewer 4 Report

The value of this paper would be further enhanced if the authors considered the following:

i)                    This paper is a review. The title should stipulate it is a review, not an investigation.

ii)                  Line 152. Persistent myelination leads to light scatter and renders the immediate tissue opaque. This does not directly influence refractive error. Please correct.

iii)                Repetition. Current content of the paper should be checked for duplicity and trimmed accordingly. Lines 65-69 are nearly identical to lines 164-169.   

iv)                In several regions the main reason for induced corneal epithelial removal is some kind of photorefractive keratectomy or corneal cross-linking. The authors should include this in the review and review the typical post-treatment regimens implemented to facilitate re-epithelization, and the mechanisms that lead to a reversal of the initial depreciation in corneal sensitivity.  

Author Response

We thank the reviewer for the valuable comments. Below are per point responses:

i) A word “Review” is present above the title

ii) The statement is marginal and is not in major focus of the manuscript, therefore, was removed.

iii) The lines 164-169 of the original text were removed

iv) The suggested paragraph was added.

Round 2

Reviewer 1 Report

Konstantin Feinberg presented an overview of the sensory innervation in homeostatic and injury-induced corneal epithelium. It's worth to be published in IJMS.

While, I have some minor sugestions:

1. The structure of the MS should be re-organized: 1) the Introduction should be just a introduction about the definition sensory innervation, homeostasis of corneal epithlium, sensory innervation related corneal epithlium destruction and provide the problems or viewpoint of the MS; 2)the part "Schwann cells and innervation-dependent epithelial renewal" should be before the therapeutic part. 

2. The References 1 seemed to be lost its way in the MS.

Author Response

We thank the reviewers for valuable comments. The revised manuscript has been reorganized according to the reviewer's suggestions. Relocated/modified parts appear as tracked changes. The reference 1 issue has been resolved.

Regards, Konstantin Feinberg